# HashOrder: Accelerating Graph Processing Through Hashing-based Reordering

## Abstract

Graph processing systems are a fundamental tool across various domains such as machine learning, and their efficiency has become increasingly crucial due to the rapid growth in data volume. A major bottleneck in graph processing systems is poor cache utilization. Graph reordering techniques can mitigate this bottleneck and significantly speed up graph workloads by improving the data locality of the graph memory layout. However, since existing approaches use greedy algorithms or simple heuristics to find good orderings, they suffer from either high computational overhead or suboptimal ordering quality. To this end, we propose HashOrder, a probabilistic algorithm for graph reordering based on randomized hashing. We theoretically show that hashing-based orderings have quality guarantees under reasonable assumptions. HashOrder produces high-quality orderings while being lightweight and parallelizable. We empirically show that HashOrder beats the efficiency-quality tradeoff curve of existing algorithms. Evaluations on various graph processing workloads and GNN data loaders reveal that HashOrder is competitive with or outperforms the existing best method while being $592\times$ more efficient in reordering, speeding up PageRank by $1.44\times$ and GNN data loaders by $1.93\times$ on average.

## 1 Introduction

Graph structured data are widely prevalent across different domains, such as search engine (Page et al., 1998), social network (Ching et al., 2015), and bioinformatics (Aittokallio & Schwikowski, 2006). As a result, graph processing systems have become an indispensable tool for analyzing and extracting insights from graph data in various fields. The rapid growth in data volume over the past years has made graphs larger and more difficult to manage (Malewicz et al., 2010). Graphs in real-world applications contain up to billions of nodes and trillions of edges (Ching et al., 2015), making the efficiency of graph processing systems increasingly crucial.

In recent years, the field of machine learning has experienced a surge in algorithmic advances for graph analysis and graph learning, e.g. in community detection (Kozdoba & Mannor, 2015), graph clustering (Korlakai Vinayak et al., 2014), graph edit distance (Ranjan et al., 2022), and node classification (Kipf & Welling, 2016). Moreover, the graph data structure is applied to solve more machine learning problems, such as HNSW for near-neighbor search (Malkov & Yashunin, 2018), decision graph for interpretable decision process (Zhu & Shoaran, 2021), and knowledge graph for natural language processing (Schneider et al., 2022). As machine learning problems continue to expand in scale and complexity, the efficiency of processing graphs has emerged as a significant challenge.

**Graph systems are bottlenecked by poor cache utilization.** Prior works (Wei et al., 2016; Zhang et al., 2017) have found that, in the latest graph processing frameworks, up to 90% of CPU cycles are wasted due to memory access stalling due to a combination of factors. Typical graph algorithms perform little computation per memory access (Zhang et al., 2017), so the total runtime is dominated by memory access time. Furthermore, graph data are large and they are usually arranged or ordered in a poor way that obstruct cache utilization. Graph processing systems can be significantly sped up by rearranging the memory layout of the graph representation (Balaji & Lucia, 2018).

Recently, attention has been drawn to the issue of poor cache utilization in graph learning applications (Bai et al., 2021; Lin et al., 2020; Zeng et al., 2021b). In graph learning, data loaders leverage graph sampling techniques to construct mini-batches for feed-forward and back-propagation on

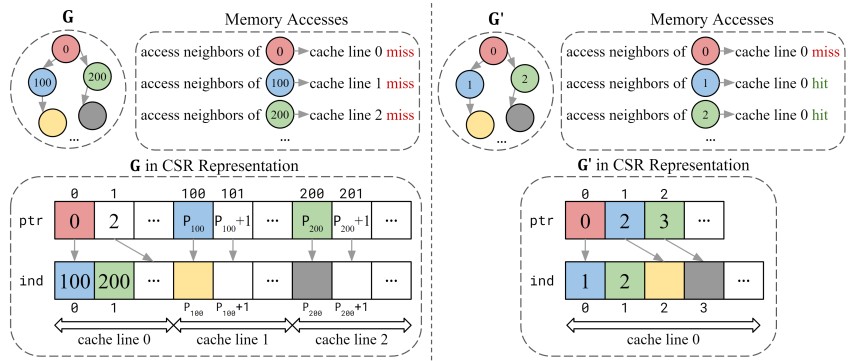

Figure 1: Illustration of how graph reordering can reduce the number of cache misses. **G** and **G′** are the same graph with different node orderings, leading to different in-memory representations. We examine the memory access patterns of breadth-first search starting from node 0. [Left] An example of a suboptimal graph ordering, which results in a high number of cache misses. [Right] A better ordering of the same graph leads to improved cache utilization.

GPU. Prior works (Lin et al., 2020; Bai et al., 2021) have observed that up to 74% of the end-to-end training time is consumed by the data loading phase, during which the GPU idles. Many graph sampling techniques, such as $k$-hop uniform neighbor sampling, are implemented with graph traversal algorithms, so they also suffer from poor cache utilization.

**Graph reordering improves cache efficiency.** Graph reordering is a general, task-agnostic approach for speeding up graph processing systems. Graph algorithms have a common memory access pattern: after a node is processed, its neighbors are processed next (Wei et al., 2016). Therefore, cache utilization can be improved by placing neighbors that are frequently co-accessed together close in memory, which translates to significant efficiency gains. Figure 1 shows an example of how graph reordering can improve cache utilization. **G** and **G′** are the same graph with distinct Compressed Sparse Row (CSR) representations due to different orderings, and they incur different memory access overhead. This memory access pattern can be generalized to a variety of graph algorithms, such as PageRank (Page et al., 1998), Bellman-Ford (Bellman, 1958), Radii, as well as graph sampling methods used in GNN training, such as uniform neighbor sampling (Hamilton et al., 2017) and GraphSAINT (Zeng et al., 2019).

**Tradeoff between reordering quality and efficiency** Due to the NP-hard nature of the graph reordering problem, the existing best algorithms employ objective-based greedy approaches (Wei et al., 2016). The greedy approaches attempt to optimize the objective in a limited local search space, and does not take the macro graph structure into account. Hence we argue that a probabilistic approach is needed for high-quality reordering by taking the higher order neighborhood into account, while keeping the computational overhead low. The existing reordering algorithms either incur significant amount of reordering time, or produce limited application speedup. We empirically study this tradeoff, and observe a general trend between reordering quality and reordering time among existing graph reordering algorithms. Figure 2 presents the reordering quality (measured by workload speedup) versus reordering time for each algorithm. The results are averaged over 5 graph workloads and 6 datasets, and the experimental setup is presented in Section 4.1. The red dotted line shows the linear gains in application speedup with exponential increase in reordering time among existing algorithms. Our proposed reordering algorithm HashOrder surpasses existing methods in terms of reordering quality, and reduces the computational overhead for achieving the best quality ordering by $592\times$ on average.

In this work, we explore a graph reordering approach based on randomized hashing. Hashing has many desirable properties in favor of the graph reordering problem. First, the efficiency of hashing allows scaling reordering to higher order neighborhoods and very large graphs. Second, the kernel of collision probability of locality-based hashing approaches are well-suited for clustering frequently co-accessed nodes together. Lastly, hashing-based reordering can be extended to handle online or streaming graphs, which is infeasible for existing objective-based reordering algorithms. We summarize the contributions of this work as follows.

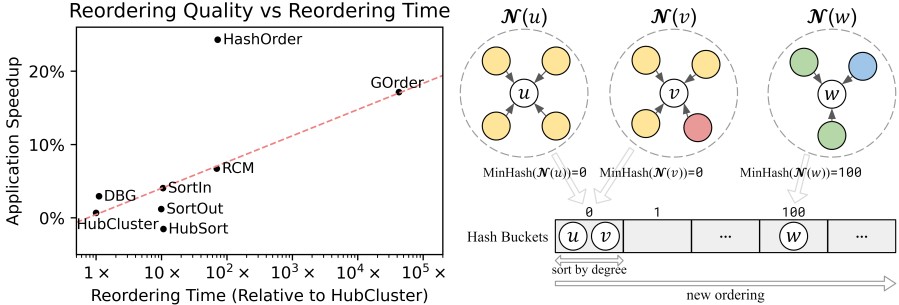

Figure 2: [Left] Tradeoff between reordering quality and reordering efficiency for different graph reordering algorithms. The red dotted line shows the growth trend of application speedup with reordering time among existing graph reordering methods. Our proposed method HashOrder surpasses all in reordering quality, while being $592\times$ more efficient in reordering than the existing best algorithm. [Right] Illustration of the HashOrder algorithm for computing graph ordering. We perform minwise hashing on the multi-hop neighborhoods of nodes to cluster frequently co-accessed nodes into hash buckets, and apply further local reordering to boost the ordering quality.

1. Propose an efficient, high-quality graph reordering algorithm based on randomized hashing. We theoretically show that the orderings computed by our algorithm has quality guarantees under reasonable assumptions.
2. Introduce a novel probabilistic perspective to the graph reordering problem. Prior algorithms employ greedy or heuristic-based approaches. We approach the problem from a probabilistic angle, and demonstrate its quality and efficiency advantages over greedy ones.
3. Present comprehensive experimental evaluations of our proposed method against competitive baselines. We perform experiments on a set of representative graph processing workloads and real-world datasets. Additionally, we demonstrate the potential of graph reordering in speeding up graph learning applications.

## 2   RELATED WORKS

**Efficient Graph Systems** There has been substantial interest in the development of efficient graph processing systems. Systems such as Pregel (Malewicz et al., 2010), GraphX (Gonzalez et al., 2014), PowerGraph (Gonzalez et al., 2012), and Ligra (Shun & Blelloch, 2013) are developed to tackle distributed or parallel processing of large-scale graphs. Locality-sensitive hashing (LSH) (Indyk & Motwani, 1998) has been applied to improve system efficiency in previous works. For example, Jiang et al. (2020) leverage LSH for efficient sparse-dense matrix multiplication, and Huang et al. (2021) uses LSH for locality-aware task scheduling in GNN. However, to the best of our knowledge, probabilistic algorithms have not been applied to the graph reordering problem.

**Graph Reordering** Low cache utilization has been identified as one of the main bottlenecks of graph processing systems (Zhang et al., 2017), and graph reordering techniques have been shown to be effective in speeding up graph systems (Balaji & Lucia, 2018; Faldu et al., 2019; Wei et al., 2016) regardless of graph workloads. GOrder (Wei et al., 2016) greedily optimizes the neighborhood overlap between nodes within a sliding window, which has been shown to be the best existing reordering method (Faldu et al., 2019; Coleman et al., 2022), but its overhead makes it impractical in many applications (Balaji & Lucia, 2018). Originally developed for matrix bandwidth reduction, Reverse Cuthill-McKee (RCM) (Cuthill & McKee, 1969) computes graph ordering by aiming to minimize the maximum label difference (Auroux et al., 2015). MLOGA and MLINA (Chierichetti et al., 2009) perform reordering by minimizing the label discrepancies between neighbors. Due to the high overhead of objective-based methods, many lightweight algorithms have been proposed. DBG (Faldu et al., 2019) and degree sorting only use the degree information of nodes to determine ordering. HubCluster (Balaji & Lucia, 2018) and HubSort (Zhang et al., 2017) are lightweight reordering methods that group hub and non-hub nodes separately. Rabbit Order (Arai et al., 2016) is a lightweight method based on clustering of hierarchical communities and can be parallelized with careful concurrency control. In this work, we choose a competitive set of baselines for evaluation, including objective-based reordering methods such as GOrder (Wei et al., 2016) and RCM (Cuthill & McKee, 1969), and a number of lightweight reordering methods.

# 3 HASHORDER: HASHING-BASED PROBABILISTIC GRAPH REORDERING

In this section, we introduce our proposed probabilistic reordering algorithm HashOrder. We first introduce the background including the problem statement and hashing schemes. Then we provide the details of our proposed algorithm. Lastly, we perform an analysis on the quality of the orderings produced by HashOrder and its complexity.

## 3.1 BACKGROUND

Given a graph $\mathcal{G} = (\mathcal{V}, \mathcal{E})$ where $|\mathcal{V}| = N$ and $|\mathcal{E}| = M$, the graph reordering problem aims to find a one-to-one node-labeling mapping $P : \mathcal{V} \to [N]$ where $[N] = \{1, \ldots, N\}$. The order defined by $P$ is used as the order in which nodes are stored in memory. The objective is to find the optimal $P$ to maximize the cache efficiency of the graph memory layout. Specifically, the fitness of the ordering $P$ can be measured as

$$\text{fit}(P) = \sum_{u \neq v \ u,v \in \mathcal{V}} S(u,v) \mathbb{1}\left( \lfloor \frac{P(u)}{B} \rfloor = \lfloor \frac{P(v)}{B} \rfloor \right) \tag{1}$$

where $S(u, v)$ measures the locality benefits of placing nodes $u, v$ within the same cache line/page, $B$ denotes the number of nodes a cache line/page can hold, and the indicator function indicates whether nodes $u, v$ are stored in the same cache line/page. Our definition of the fitness of an ordering is slightly different from that of Wei et al. (2016), which considers the sum of $S(u, v)$ within a sliding window. We note that our definition more closely reflects cache efficiency on the hardware level, since cache lines/pages do not overlap with one another. We define the graph ordering problem as follows.

**Definition 1** (Graph ordering($\mathcal{G}$)). *Given a graph $\mathcal{G} = (\mathcal{V}, \mathcal{E})$, the problem of graph ordering is to find an ordering $P^\star : \mathcal{V} \to [N]$ such that*

$$P^\star = \arg\max_P \text{fit}(P) \tag{2}$$

Prior works such as Wei et al. (2016) have used the number of common in-neighbors (including self-loop) to measure the locality benefits of collocation of two nodes, i.e., $S(u, v) = \mathcal{N}_{\text{in}}^+(u) \cap \mathcal{N}_{\text{in}}^+(v)$, where $\mathcal{N}_{\text{in}}^+(u) = \{v \mid (v, u) \in \mathcal{E}\} \cup \{u\}$, since the out-neighbors of a node are likely to be processed together. We note that this is just one possible definition of $S(u, v)$, and we use a similar definition which is more amenable to optimizing with hash functions. We leverage Locality-sensitive Hashing (LSH) (Indyk & Motwani, 1998) to quickly and effectively optimize the fitness of an ordering. Formally, LSH is defined as follows.

**Definition 2** (Locality-sensitive hashing). *A hash family $\mathcal{H}$ is $(R_1, R_2, p_1, p_2)$-sensitive with respect to a pairwise similarity function $S(\cdot, \cdot)$ if the following conditions holds for all points $a, b$,*

$$S(a,b) \geq R_1 \implies \Pr_{h \leftarrow \mathcal{H}} \big( h(a) = h(b) \big) \geq p_1 \text{ and } S(a,b) \leq R_2 \implies \Pr_{h \leftarrow \mathcal{H}} \big( h(a) = h(b) \big) \leq p_2 \tag{3}$$

It has been shown by Coleman et al. (2022) that using the size of the in-neighborhood intersection as the optimization objective has direct connections to the cache efficiency metric in the ideal cache model. Therefore, it is reasonable to use an LSH function which operates on the in-neighborhood of an input node, and outputs hash codes with collision probability correlated with a similarity measure of the in-neighborhoods. Possible options include SimHash for cosine similarity, MinHash for Jaccard similarity, and more. We choose MinHash for its proven robust performance on sets (Shrivastava & Li, 2014). We present our proposed algorithm in the next section.

## 3.2 OUR PROPOSAL

Our insight is that we can leverage probabilistic algorithms to quickly and effectively optimize the fitness of an ordering by clustering nodes with similar neighborhoods, then apply lightweight local reordering methods within each cluster to boost ordering quality. Specifically, we first leverage MinHash to compute data-oblivious locality-sensitive hash codes, and group nodes into locality-aware clusters for local processing. Within each cluster of nodes, we perform independent reordering through neighbor grouping or degree sorting. Pseudocode for the full algorithm is provided in Algorithm 1. We provide details about the design of our algorithm in the following paragraphs.

---

**Algorithm 1** HashOrder algorithm for efficient computation of high-quality graph ordering

---

1: **procedure** HASHORDER($\mathcal{G}$, hops, $l$)
2: **input** Graph $\mathcal{G} = (\mathcal{V}, \mathcal{E})$, number of hops hops, number of hash functions $l$
3: **output** Ordering $P : \mathcal{V} \rightarrow [N]$
4:      initialize $l$ random permutations $\pi_1, \ldots, \pi_l : \mathcal{V} \rightarrow [N]$
5:      let $h_0^j[v] \leftarrow \pi_j[v]$ for all $j \in [l], v \in \mathcal{V}$
6:      **for** $i \leftarrow 1 \ldots$ hops **do**             ▷ multi-hop hash codes computation via message passing
7:          **for** $v \in \mathcal{V}$ **do**
8:              let $h_i^j[v] \leftarrow \min_{(u,v) \in \mathcal{E} \cup (v,v)} h_i^{j-1}[u]$ for all $j \in [l]$
9:          **end for**
10:     **end for**
11:      let $h[v] \leftarrow \text{concat}(h_{\text{hops}}^1[v], \ldots, h_{\text{hops}}^l[v])$ for all $v \in \mathcal{V}$
12:      let $p \leftarrow$ INBUCKETORDER($\mathcal{G}, h$)         ▷ apply neighbor grouping or degree sorting
13:      let $P[p[i]] \leftarrow i$ for $i \in [N]$
14:      **return** $P$
15: **end procedure**

16: **procedure** NEIGHBORGROUPINGORDER($\mathcal{G}, h$)
17: **input** Graph $\mathcal{G} = (\mathcal{V}, \mathcal{E})$, hash codes $h : \mathcal{V} \rightarrow [N]$
18: **output** Inverse ordering $p : [N] \rightarrow \mathcal{V}$
19:      let $O : \mathcal{V} \rightarrow [N]$ be the order of breadth-first traversal starting from a random node
20:      let $Q[v] \leftarrow (h[v], O[v])$ for all $v \in \mathcal{V}$
21:      let $p \leftarrow \text{argsort}(Q)$
22:      **return** $p$
23: **end procedure**

24: **procedure** DEGREESORTINGORDER($\mathcal{G}, h$)
25: **input** Graph $\mathcal{G} = (\mathcal{V}, \mathcal{E})$, hash codes $h : \mathcal{V} \rightarrow [N]$
26: **output** Inverse ordering $p : [N] \rightarrow \mathcal{V}$
27:      let $Q[v] \leftarrow (h[v], \text{degree}(v))$ for all $v \in \mathcal{V}$
28:      let $p \leftarrow \text{argsort}(Q)$
29:      **return** $p$
30: **end procedure**

---

**MinHash** Min-wise independent permutations (MinHash) (Broder, 1997) is a family of randomized hash functions designed for estimating set resemblance. Let $U$ be a set and $X \subseteq U$, and $\pi$ be a random permutation over $U$. The MinHash function $h_{\min}$ is defined as follows

$$h_{\min}(X) = \min_{x \in X} \pi(X) \tag{4}$$

The collision probability for $X_1, X_2 \subseteq U$ under MinHash is the Jaccard similarity between them.

$$\mathbf{Pr}(h_{\min}(X_1) = h_{\min}(X_2)) = |X_1 \cap X_2| / |X_1 \cup X_2| \tag{5}$$

We use the multi-hop neighborhood of nodes as the input to MinHash to obtain clusters of frequently co-accessed nodes, and apply fine-grained local reordering methods independently on each cluster to determine the final ordering.

**Multi-hop neighborhood** Previous works have only considered single-hop neighborhood for computing graph ordering (Wei et al., 2016; Cuthill & McKee, 1969). We observe empirically that using multi-hop neighborhood produces better ordering quality, as shown in Figure 5. It is computationally expensive for previous reordering methods such as GOrder (Wei et al., 2016) to scale to multi-hop neighborhoods due to the exponential growth of neighborhood size with the number of hops. However, our algorithm circumvents this problem by computing hash codes of multi-hop neighborhood through message passing, which scales linearly with the number of hops. An analysis on time complexity is given in Section 3.3.

**Neighbor grouping or degree sorting within buckets** After nodes with the same hash code have been grouped into buckets, we apply local reordering within buckets to further improve ordering quality. We consider two lightweight reordering techniques: neighbor grouping and degree sorting.

Neighbor grouping traverses the graph in a breadth-first manner, and the order of traversal becomes the ordering of the nodes. This is motivated by the common memory access pattern of graph algorithms; since neighbors of a node are likely to be accessed together, they should be grouped together to improve layout locality. On the other hand, degree sorting computes the ascending in-degree order of nodes, placing frequently accessed nodes together. Empirically, we observe marginally better ordering quality from neighbor grouping over degree sorting, at the cost of higher reordering time. We present a empirical comparison on in-bucket reordering techniques in Figure 5.

**Parallelizability** HashOrder can be made massively parallel without locking or thread communication. The work of hash code computation can be divided evenly among available threads, while in-bucket reordering techniques can be computed independently in parallel for each bucket. The sorting operation can be done with a parallel sorting algorithm. The parallelizability of HashOrder is an advantage over previous methods such as GOrder (Wei et al., 2016) and RCM (Cuthill & McKee, 1969), which have no known parallel version due to their greedy sequential computation.

## 3.3 ANALYSIS

In this section, we perform an analysis on the quality of our proposed hashing-based ordering. We also analyze the time complexity of our algorithm. First, we define the concept of hashing-induced ordering to formally define how graph orderings can be derived from a hash function.

**Definition 3** (Hashing-induced ordering). *Let $h \in \mathcal{H} : V \to [R]$ be a hash function that maps nodes to the range $[R]$. Then $P_h$ is an $h$-induced ordering if the following holds for all $u, v \in \mathcal{V}$,*

$$h(u) < h(v) \implies P_h(u) < P_h(v) \tag{6}$$

Put simply, a graph ordering is considered $h$-induced if the nodes are ordered in ascending order of their hash codes from $h(\cdot)$, regardless of tie-breaking rules. Algorithm 1 produces valid hashing-induced orderings. Next, we define graph clusters and separable clusters, which are assumptions of the underlying graph structure for inferring the quality of hashing-induced orderings.

**Definition 4** (($S, \alpha, \beta$) clustering). *Given a graph $\mathcal{G} = (\mathcal{V}, \mathcal{E})$, then the disjoint partitions of the nodes, denoted by $\{\mathcal{V}_1, \ldots, \mathcal{V}_C\}$, is a $(\alpha, \beta)$ clustering with respect to a similarity $S$ if the following holds,*

$$\forall i, j \in [C], i \neq j, \forall v_a \in \mathcal{V}_i, \forall v_b \in \mathcal{V}_j, \; S(v_a, v_b) \leq \alpha \tag{7}$$

$$\forall i \in [C], \forall v_a, v_b \in \mathcal{V}_i, \; S(v_a, v_b) \geq \beta \tag{8}$$

Hash functions are capable of distinguishing graph clusters, as formalized in the following lemma.

**Lemma 1.** *Consider a graph $\mathcal{G}$ with a $(S, \alpha, \beta)$ clustering $\{\mathcal{V}_1, \ldots, \mathcal{V}_C\}$ where similarity $S$ accepts a $(R, \alpha, p_1, p_2)$ LSH family $\mathcal{H}$. Let $\mathcal{V}_a$ and $\mathcal{V}_b$ be two clusters in $\{\mathcal{V}_1, \ldots, \mathcal{V}_C\}$. Consider a hash function $h$ constructed by concatenating $l$ independently drawn hash functions from $\mathcal{H}$. If $l \geq \frac{\ln \frac{\delta}{|\mathcal{V}_a||\mathcal{V}_b|}}{\ln p_2}$, then the following holds with probability at least $(1 - \delta)$*

$$\forall v_a \in \mathcal{V}_a, v_b \in \mathcal{V}_b, \; h(v_a) \neq h(v_b) \tag{9}$$

The proof is deferred to the appendix. Lemma 1 implies that given an LSH family $\mathcal{H}$ for the similarity measure $S(\cdot, \cdot)$, we can construct a hash function by concatenating hash functions drawn from $\mathcal{H}$, such that nodes from different clusters are placed into different buckets with arbitrarily high probability. For Jaccard Similarity $\mathcal{J}$, which we will be using in our algorithm, for any given $\alpha$ and $R$ such that $R > \alpha$, MinHash is a $(R, \alpha, R, \alpha)$ sensitive family. Our objective is to find a graph ordering that maximizes the fitness. With reasonable assumption on the graph, we can construct a hash function such that the fitness of its induced ordering is lower bounded by a constant factor of the optimal fitness, formalized as follows.

**Theorem 1.** *Consider a graph $\mathcal{G}$ with a $(S, \alpha, \beta)$ clustering $\{\mathcal{V}_1, \ldots, \mathcal{V}_C\}$ with size $C$ where similarity $S$ accepts a $(R, \alpha, p_1, p_2)$ LSH family $\mathcal{H}$. Suppose the number of nodes is multiple of cache-line width. Consider a hash function $h$ constructed by concatenating $l$ independently drawn hash functions from $\mathcal{H}$. For any given $\delta \in (0, 1)$, if $l \geq \frac{\ln \frac{C(C-1)\delta}{2 \max_{i,j,i \neq j} |\mathcal{V}_i||\mathcal{V}_j|}}{\ln p_2}$, then the following holds with probability at least $(1 - \delta)$*

$$\beta fit(P^\star) \leq fit(P_h) \tag{10}$$

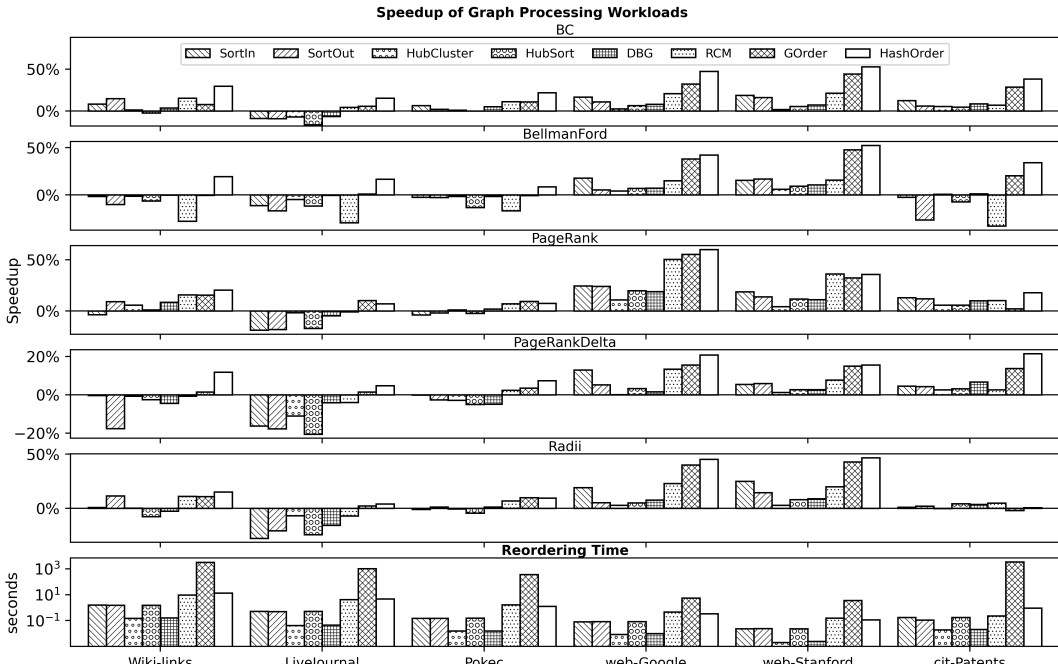

Figure 3: [Top 5] Speedup of graph processing workloads by different graph reordering algorithms. HashOrder mostly outperform baselines in reordering quality, achieving up to 59.9% speedup in PageRank. [Bottom 1] Reordering time of graph reordering algorithms for each dataset. HashOrder's reordering time is on par with RCM, and on average $592\times$ more efficient than GOrder.

The proof is provided in the appendix. This result provides a guarantee for the quality of our proposed hashing-based orderings with high probability. Furthermore, it illustrates the reasoning behind the in-bucket local ordering techniques such as neighbor grouping and degree sorting. The constant factor $\beta$, which lower bounds the fitness of hashing-induced orderings, takes into account that bucket size may exceed cache line capacity and ordering within a bucket may be sub-optimal. Therefore, by applying local reordering techniques within each bucket, we can potentially improve the fitness of a hashing-induced ordering further. Although clustering assumptions are made in our theoretical results, we observe empirically that HashOrder produces high-quality ordering across a variety of graphs.

**Time and space complexity** HashOrder has overall time complexity $O\big(k|\mathcal{E}|+|\mathcal{V}|\log|\mathcal{V}|\big)$ and space complexity $O\big(|\mathcal{V}|\big)$. The $k$-hop neighborhood hash code computation via message passing has complexity $O\big(k|\mathcal{E}|\big)$. The complexity of breadth first search and degree counting are $O\big(|\mathcal{V}|+|\mathcal{E}|\big)$ and $O\big(|\mathcal{E}|\big)$, respectively. Lastly, the complexity of the sorting operation over nodes is $O\big(|\mathcal{V}|\log|\mathcal{V}|\big)$. The algorithm uses a constant number of $|\mathcal{V}|$-sized arrays, giving a space complexity of $O\big(|\mathcal{V}|\big)$. Since message passing only depends on the hash codes of the previous hop, the space complexity does not depend upon the number of hops $k$. Empirically, the reordering speed of HashOrder is on par with RCM, and orders of magnitude faster than GOrder.

## 4  EXPERIMENTS

In this section, we present experimental evaluations to compare HashOrder and baselines for graph processing workloads and graph learning applications. We design the experiments to compare the quality and efficiency of HashOrder with competitive baselines.

### 4.1  EXPERIMENTAL SETUP

We perform experiments on two catagories of graph workloads: traditional graph processing algorithms, and data loading for graph learning.

**Evaluation Metrics** We are interested in evaluating the quality of the ordering and the efficiency of the computation for each reordering algorithm. We measure the quality of a graph ordering by

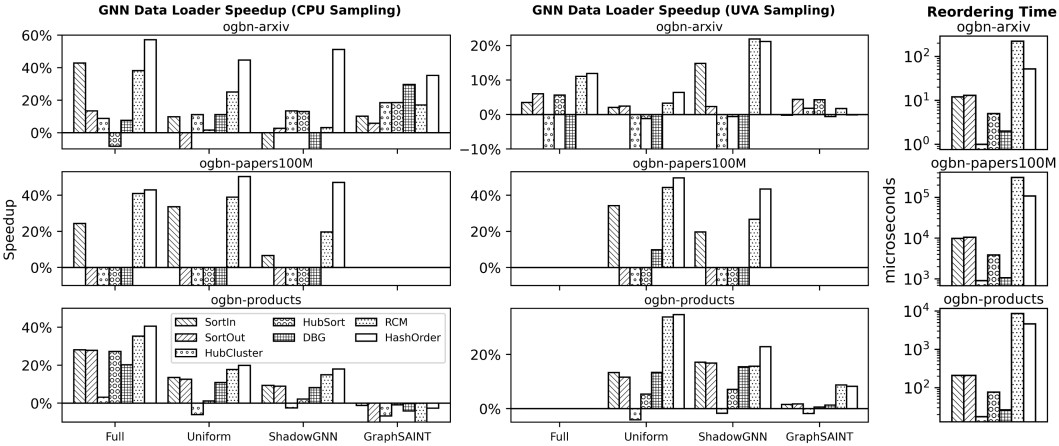

Figure 4: [Left 2] GNN data loader speedup after applying different graph reordering techniques. HashOrder mostly outperforms baselines, achieving up to 57.1% speedup in GNN loader loaders. [Right] Reordering time by each algorithm for different datasets.

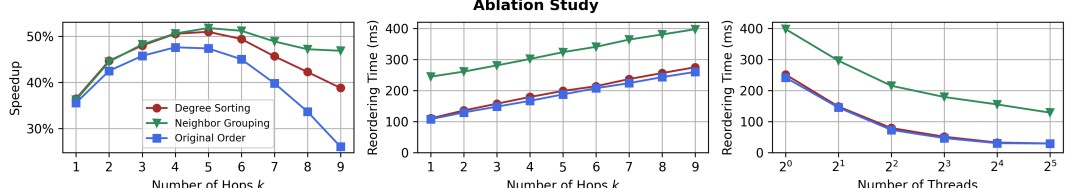

Figure 5: The results of the ablation study. [Left 2] The effects of in-bucket reordering techniques and the number of hops $k$ on the reordering quality and reordering time. [Right 1] The parallelizability of HashOrder with an increasing number of threads.

using the metric *relative speedup* $= 100\frac{t_0 - t_1}{t_0}\%$, where $t_0$ is the workload execution time on the original graph, and $t_1$ is the execution time on the reordered graph. The process is repeated 5 times independently and the results are averaged. The computational cost for computing graph orderings is measured as the running time of a reordering algorithm, using a single CPU thread.

**Baselines** We consider the following competitive graph reordering methods as baselines. 1. GOrder (Wei et al., 2016), 2. reverse Cuthill–McKee (RCM) (Cuthill & McKee, 1969), 3. Degree-based Grouping (DBG) (Faldu et al., 2019), 4. HubCluster (Balaji & Lucia, 2018), 5. HubSort (Zhang et al., 2017), 6. in-degree sorting (SortIn), 7. out-degree sorting (SortOut).

**Hyperparameters** For GOrder, we consider the best window size $w \in \{5, 10\}$. For DBG, we consider the number of groups $g = 8$. For HashOrder, we consider the best number of hops $k \in \{2, 4, 6, 8\}$ and number of hash functions $l = 2$. Other algorithms have no tunable parameters.

**Testbed** We perform experiments on a Linux system equipped with 2 Intel Xeon Gold 5220R processors (L1, L2, L3 cache sizes are 1.5MB, 48MB, and 71.5MB, respectively), 1.5T of RAM, and one NVIDIA Tesla V100 GPU.

**Software Implementation** HashOrder and baseline methods are implemented in C++ and compiled with the `-O3` flag. We use the Ligra framework (Shun & Blelloch, 2013) for running graph processing workloads. For GNN data loaders, we use Deep Graph Library (DGL) (Wang, 2019), which provides efficient implementation of graph sampling algorithms in C++ and OpenMP, and supports graph sampling using unified virtual memory (UVA).

**Graph Processing Workloads and Datasets** Commonly used graph processing workloads chosen for evaluation and their applications are as follows, 1. Betweenness Centrality (BC) for measuring graph centrality, 2. PageRank (Page et al., 1998) for search engine ranking, 3. PageRankDelta (Shun & Blelloch, 2013) for lightweight search engine ranking, 4. Radii Estimation (Radii) for graph diameter estimation, 5. BellmanFord (Bellman, 1958) for single-source shortest path computation. We choose 6 large-scale real-world graph datasets (Leskovec & Krevl, 2014) for evaluation. Statistics of the datasets are presented in the appendix.

**Data Loading for GNN** We evaluating the ability of graph reordering algorithms for speeding up GNN data loaders. We evaluate two scenarios, 1. when graph sampling is done by CPU with 1 worker, 2. when graph sampling is done by GPU using unified virtual addressing (UVA) memory (Huang et al., 2022). We perform 4 types of graph sampling algorithms 1. full neighborhood sampling (Full), 2. uniform neighbor sampling (Uniform) (Hamilton et al., 2017), 3. GraphSAINT sampling (Zeng et al., 2019), 4. ShadowGNN sampling (Zeng et al., 2021a), on 3 real-world datasets (Hu et al., 2020), whose statistics are presented in the appendix. We record the time for a GNN data loader to perform 3-hop neighborhood sampling and mini-batch construction for 100 iterations, and results are averaged over 3 independent runs.

## 4.2 EXPERIMENTAL RESULTS

**Results on Graph Processing Workloads** We evaluate the quality and efficiency of graph reordering algorithm on graph processing workloads. Figure 3 presents the reordering quality and reordering time for HashOrder and baselines on graph processing workloads. HashOrder mostly outperforms baselines in application speedup. On LiveJournal and Pokec, HashOrder is able to achieve considerable speedup while most baselines result in negative workload speedup. We highlight that HashOrder achieves up to 52.7%, 52.3%, and 59.9% speedup for BC, BellmanFord, and PageRank workloads, respectively. The comparison of reordering time, using a single thread, is shown at the bottom of Figure 3. On average, HashOrder uses $1.03\times$ reordering time for reordering as RCM, but achieves much better reordering quality. Compared with GOrder, HashOrder is $592\times$ faster in reordering on average and achieves higher workload speedup. Figure 2 presents a direct comparison of speedup versus reordering time for all methods, averaged over all datasets and workloads.

**Cache Utilization Analysis** We profile the graph workloads using the Valgrind tool to analyze the cache efficiency of graphs after reordering with each method. The results on cache miss rates are presented in the appendix. The cache miss rate is negatively correlated with application speedup, confirming that the speedup indeed comes from improved cache utilization.

**Results on GNN Data Loading** We reorder graph datasets for GNN training and measure the speedup of GNN data loaders. Figure 4 presents the results on data loader speedup after reordering the graphs with HashOrder and baselines. Some data loading configurations, e.g. UVA-based full neighborhood sampling on ogbn-papers100M, are infeasible due to CUDA or memory limitations, therefore omitted. We omit GOrder as a baseline since it fails to finish reordering ogbn-products or ogbn-papers100M in 10 hours. HashOrder outperforms baselines in almost all cases, achieving up to 57.1% speedup for GNN data loading. The rightmost plot in Figure 4 presents the reordering time for each method. HashOrder has lower computational overhead for reordering than the second best method RCM, while achieving higher ordering quality.

**Ablation Study** In this section, we perform ablation experiments to study each component of HashOrder, using the PageRank workload on the web-Google dataset. We vary the number of hops $k$ in the hashed neighborhood and change the in-bucket ordering operations to study their effects on the quality and efficiency of reordering. Figure 5 presents the results on the ablation experiments. Among all three in-bucket ordering techniques considered, neighbor grouping achieves the most speedup at the cost of higher reordering overhead. In-bucket degree sorting achieves comparable ordering quality as neighbor grouping while using much lower reordering time. On the other hand, maintaining the original order of nodes within buckets produces lower reordering quality. As we increase the number of hops $k$, the reordering quality first increases then decreases, due to the monotonic increase of the number of hash collisions with increasing $k$. Hash buckets that hold too few or too many nodes negatively affect the data locality. The reordering time increases linearly with $k$, which is consistent with our time complexity analysis. Finally, HashOrder is scalable with increasing number of CPU threads.

## 5 CONCLUSION

We propose HashOrder, an efficient and high-quality graph reordering algorithm based on randomized hashing. HashOrder achieves efficient computation and parallelizability, while achieving high ordering quality. Different from existing objective-based reordering methods, our proposed approach is a novel probabilistic perspective on the graph reordering problem. Empirical evaluations on typical graph processing workloads and data loading for GNN show that HashOrder is effective in accelerating graph systems while incurring low overheads, and it beats the efficiency-quality tradeoff curve of existing approaches.

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

## A  APPENDIX

### A.1  DATASETS

Tables 1 and 2 present the statistics of the datasets used for graph processing and GNN data loader experiments, respectively.

Table 1: Statistics of the datasets used for graph processing experiments.

| Dataset | # of Nodes | # of Edges | Type |
| --- | --- | --- | --- |
| Wiki-links (Kunegis, 2013) | 18,268,992 | 172,183,984 | Web graph |
| LiveJournal (Leskovec & Krevl, 2014) | 3,997,962 | 34,681,189 | Social network |
| cit-Patents (Leskovec & Krevl, 2014) | 3,774,768 | 16,518,948 | Citation network |
| Pokec (Leskovec & Krevl, 2014) | 1,632,803 | 30,622,564 | Social network |
| web-Google (Leskovec & Krevl, 2014) | 875,713 | 5,105,039 | Web graph |
| web-Stanford (Leskovec & Krevl, 2014) | 281,903 | 2,312,497 | Web graph |

### A.2  CACHE ANALYSIS

We profile graph workloads using the Valgrind tool to analyze the cache efficiency of the graph after reordering with each method. We measure the cache miss rate, which is defined as

$$\text{cache miss rate} = 100 \frac{\text{\# of cache misses}}{\text{\# of memory accesses}}\%  \tag{11}$$

The cache miss rates on the web-Google and web-Stanford dataset are given in Tables 3 and 4, respectively. HashOrder achieves the lowest L3 cache miss rates in all cases, and the lowest L1 cache miss rates in most of the cases. Since the L3 cache is orders of magnitude slower than L1, a reduction in L3 cache miss rate leads to much more application speedup than the same reduction in L1 cache miss rate. The reductions in cache miss rate are highly correlated with the workload speedup, proving that the application speedup comes from improved cache efficiency.

Table 2: Statistics of the datasets used for GNN data loader experiments.

| Dataset | # of Nodes | # of Edges | Type |
|---|---|---|---|
| ogbn-arxiv (Hu et al., 2020) | 169,343 | 1,166,243 | Citation network |
| ogbn-products (Hu et al., 2020) | 2,449,029 | 61,859,140 | Co-purchase network |
| ogbn-papers100M (Hu et al., 2020) | 111,059,956 | 1,615,685,872 | Citation network |

Table 3: The L1/L3 cache miss rates of graph workloads on the web-Google dataset for each graph reordering technique. HashOrder achieves the lowest L3 cache miss rates in all cases, leading to the most workload speedup.

| | BC | | BellmanFord | | PageRank | | PageRankDelta | | Radii | |
|---|---|---|---|---|---|---|---|---|---|---|
| | L1 | L3 | L1 | L3 | L1 | L3 | L1 | L3 | L1 | L3 |
| Original | 17.68% | 5.18% | 10.04% | 2.80% | 28.35% | 5.73% | 6.66% | 1.85% | 23.50% | 4.01% |
| HubCluster | 13.59% | 3.80% | 8.62% | 2.69% | 27.76% | 4.39% | 6.29% | 1.69% | 21.04% | 3.54% |
| HubSort | 11.66% | 3.81% | 7.98% | 2.67% | 27.36% | 4.35% | 5.91% | 1.70% | 19.50% | 3.56% |
| DBG | 12.35% | 3.74% | 8.29% | 2.65% | 27.56% | 4.02% | 6.12% | 1.69% | 20.69% | 3.55% |
| SortOut | 12.99% | 3.79% | 8.49% | 2.68% | 27.30% | 3.74% | 6.19% | 1.68% | 19.96% | 3.53% |
| SortIn | 11.31% | 3.52% | 7.86% | 2.48% | 26.84% | 4.31% | 5.80% | 1.58% | 19.20% | 3.34% |
| RCM | 8.57% | 3.37% | 6.21% | 2.42% | 9.77% | 3.37% | 4.42% | 1.54% | 11.17% | 3.40% |
| GOrder | **5.73%** | 2.88% | 4.34% | 1.83% | 6.84% | 3.49% | 3.46% | 1.49% | **5.45%** | 3.21% |
| HashOrder | 6.38% | **2.60%** | **4.12%** | **1.78%** | **6.72%** | **3.15%** | **3.43%** | **1.45%** | 5.91% | **3.04%** |

## A.3 PROOFS

In this section, we present proofs to the lemmas and theorems presented in the main paper. First, we prove Lemma 1 as follows.

*Proof.* We prove the existence of such a hash function by constructing one that satisfies the conditions. If we construct a hash function $h(\cdot)$ by concatenating $l$ hash functions $h_1(\cdot), \ldots, h_l(\cdot)$ independently drawn from the $(R, \alpha, p_1, p_2)$-sensitive hash family, where $l \geq \frac{\ln \frac{\delta}{|\mathcal{V}_1||\mathcal{V}_2|}}{\ln p_2}$, then

$$\Pr\left(\bigcup_{v_1 \in \mathcal{V}_1, v_2 \in \mathcal{V}_2} h(v_1) = h(v_2)\right) \leq \sum_{v_1 \in \mathcal{V}_1, v_2 \in \mathcal{V}_2} \Pr(h(v_1) = h(v_2)) \tag{12}$$

$$= \sum_{v_1 \in \mathcal{V}_1, v_2 \in \mathcal{V}_2} \prod_{i \in [l]} \Pr(h_i(v_1) = h_i(v_2)) \tag{13}$$

$$\leq |\mathcal{V}_1||\mathcal{V}_2|p_2^l \tag{14}$$

$$\leq \delta \tag{15}$$

$\square$

The proof for Theorem **??** is provided as follows.

*Proof.* For each cluster $\mathcal{G}_i, i \in [C]$, let $(1 - \epsilon)s_i \leq S(u, v) \leq (1 + \epsilon)s_i$ for all $u, v \in \mathcal{V}_i$, due to the property of graph cluster. Then, because of $\alpha$-separability of clusters, the fitness of the optimal ordering $P^\star$ is upper bounded by the sum of maximum $S(u, v)$ within a cache line/page over all clusters,

$$\text{fit}(P^\star) \leq \sum_{i=1}^{C} \sum_{j=1}^{|\mathcal{V}_i|/B} \frac{B(B-1)}{2}(1 + \epsilon)s_i \tag{16}$$

Table 4: The L1/L3 cache miss rates of graph workloads on the web-Stanford dataset for each graph reordering technique.

| | BC | | BellmanFord | | PageRank | | PageRankDelta | | Radii | |
|---|---|---|---|---|---|---|---|---|---|---|
| | L1 | L3 | L1 | L3 | L1 | L3 | L1 | L3 | L1 | L3 |
| Original | 16.04% | 0.74% | 9.12% | 1.16% | 27.81% | 1.00% | 6.04% | 0.64% | 22.35% | 1.31% |
| HubCluster | 11.87% | 0.75% | 8.04% | 1.13% | 26.33% | 1.02% | 5.64% | 0.63% | 20.97% | 1.31% |
| HubSort | 10.66% | 0.74% | 7.30% | 1.13% | 22.84% | 1.00% | 5.01% | 0.62% | 18.61% | 1.31% |
| DBG | 10.86% | 0.74% | 7.44% | 1.14% | 24.24% | 1.00% | 5.16% | 0.63% | 19.85% | 1.29% |
| SortOut | 9.43% | 0.73% | 7.14% | 1.10% | 21.80% | 0.99% | 4.94% | 0.60% | 14.00% | 1.21% |
| SortIn | 9.63% | 0.72% | 7.04% | 1.06% | 24.31% | 1.01% | 4.87% | 0.58% | 16.62% | 1.21% |
| RCM | 7.75% | 0.71% | 6.10% | 1.02% | 8.85% | 0.95% | 4.16% | 0.57% | 10.95% | 1.13% |
| GOrder | **4.97%** | 0.69% | **4.19%** | 0.97% | 9.13% | 0.98% | **3.06%** | 0.55% | 5.23% | 0.90% |
| HashOrder | **4.97%** | **0.68%** | 4.21% | **0.94%** | **8.60%** | **0.94%** | 3.15% | **0.52%** | **5.17%** | **0.83%** |

By Lemma 1, we have the existence of a hash function $h$ that assigns non-overlapping hash codes to different clusters. Since the number of nodes in each bucket is assumed to be divisible by $B$, we have

$$\mathrm{fit}(P_h) \geq \sum_{i=1}^{C} \sum_{j=1}^{|\mathcal{V}_i|/B} \frac{B(B-1)}{2}(1-\epsilon)s_i \tag{17}$$

$$\frac{1+\epsilon}{1-\epsilon}\mathrm{fit}(P_h) \geq \sum_{i=1}^{C} \sum_{j=1}^{|\mathcal{V}_i|/B} \frac{B(B-1)}{2}(1+\epsilon)s_i \tag{18}$$

$$\frac{1+\epsilon}{1-\epsilon}\mathrm{fit}(P_h) \geq \mathrm{fit}(P^\star) \tag{19}$$

$\square$

## A.4 REPRODUCIBILITY

The anonymized repository of the source code can be found at `anonymous.4open.science/r/hashorder`.

