# OpenReview forum: "HashOrder: Accelerating Graph Processing Through Hashing-based Reordering"
_ICLR.cc/2024/Conference — Submitted to ICLR 2024_

### Official Review · Reviewer_n1d2 · 2023-10-25

**Soundness:** 3 good
**Presentation:** 3 good
**Contribution:** 3 good
**Rating:** 8
**Confidence:** 4

**Summary:**

Graph reordering is effective in improving the cache utilization of graph processing. This paper observes that existing graph reordering algorithms have a tradeoff between effectiveness (i.e., quality of the reordering) and efficiency (i.e., execution time of the reordering algorithm). To mitigate this tradeoff, the paper proposes a new graph reordering algorithm that uses MinHash to generate hash signatures for the nodes and then sorts the signatures for reordering. Experiment results show that the HashOrder algorithm is both efficient and effective.

**Strengths:**

I enjoy this paper because graph reordering is a well-known difficult problem, and the paper solves it with a novel and effective idea.
1. Using MinHash to conduct graph reordering is a novel idea and makes sense.
2. Section 3.3 shows why MinHash works in theory.
3. The experiment results are strong, showing that the proposed HashOrder algorithm is efficient and effective.

**Weaknesses:**

The authors may be more detailed when discussing how hashing is used to handle graph problems in the related work, e.g., by describing how a graph problem is mapped to hashing. This does not hurt the novelty of the paper and allows readers to learn more.

**Questions:**

A common trick in hashing is to hash multiple times to enlarge the collision probability gap between similar and dissimilar object pairs. Any idea on how HashOrder can benefit from multiple hash signatures? I think that may be an interesting problem.

---

> ### Comment · Reviewer_n1d2 · 2023-11-22
> **After response**
>
> I am largely positive for the paper but surprised that the authors do not provide responses to the reviews. As such, I will not fight against rejecting the paper.

---

> ### Author Response · Authors · 2023-11-22
>
> We apologize for not posting the responses earlier. We had an unusually large number of reviewers and we had been working on a large set of additional experiments to address the reviewers' concerns. We will be posting the responses shortly.

---

> ### Author Response · Authors · 2023-11-23
>
> We thank the reviewer for recognizing the strengths of the paper in terms of the novelty of the idea and effectiveness of the method. We respond to the reviewer’s comments as follows.
>
> **1. Discussion on how hashing is used to handle graph problems in the related work.**
>
> Some notable related works include [1, 2, 3]. Specifically, [1] develops a highly efficient graph community detection method based on MinHash, [2] derives meaningful features from MinHash and HyperLogLog for learning link prediction models on graphs, [3] develops a scalable set-based graph summarization algorithm based on LSH. We will add the discussion to the final paper.
>
> **2. Any ideas on how HashOrder can benefit from multiple hash signatures?**
>
> Our final hash function is indeed composed of the concatenation of multiple hash functions. By increasing the number of hash functions, we can ensure that the hash function can distinguish whether a pair of nodes lie inside a cluster or in different clusters. The exact characterization of this property and required number of hash functions to obtain theoretical guarantees is provided in Theorem 1. Empirically, we found l=2 improves the ordering quality over l=1 (0.27% and 0.19% higher relative speedup), where l is the number of hash signatures. l=4 negatively impacts the ordering quality (7.89% and 1.46% lower relative speedup than 2 hashes), likely due to diminishing probability of collision for pairs inside a cluster.
>
> | PageRank run time in seconds (relative speedup) |                 |                 |                 |
> |-------------------------------------------------|-----------------|-----------------|-----------------|
> | Dataset                                         | l=1             | l=2             | l=4             |
> | web-Google                                      | 2.706 (57.68\%) | 2.689 (57.95\%) | 3.197 (50.06\%) |
> | web-Stanford                                    | 0.830 (39.36\%) | 0.827 (39.55\%) | 0.847 (38.09\%) |
>
> References
>
> [1] Chamberlain, Benjamin Paul, et al. "Real-time community detection in full social networks on a laptop." PloS one 13.1 (2018): e0188702.
>
> [2] Chamberlain, Benjamin Paul, et al. "Graph neural networks for link prediction with subgraph sketching." arXiv preprint arXiv:2209.15486 (2022).
>
> [3] Yong, Quinton, et al. "Efficient graph summarization using weighted lsh at billion-scale." Proceedings of the 2021 International Conference on Management of Data. 2021.

---

### Official Review · Reviewer_EBBV · 2023-10-26

**Soundness:** 2 fair
**Presentation:** 3 good
**Contribution:** 1 poor
**Rating:** 3
**Confidence:** 4

**Summary:**

This paper emphasizes the importance of improving cache utilization in efficiently processing large graphs. To address low cache utilization in graph learning and other applications, this paper studies graph reordering, a task-independent approach to improving cache efficiency, and proposes HashOrder, a probabilistic algorithm using randomized hashing. The authors try to improve the efficiency of memory access. The authors also discuss the tradeoff between the quality of graph reordering and the efficiency of the reordering process and argue that the proposed algorithm, HashOrder, improves the quality of reordering while significantly reducing the computational overhead. They also show that the proposed hashing-based ordering guarantees a certain quality, assuming the cache efficiency is based on a specific fitness. Experimental comparison with existing methods confirms that a certain speedup is achieved for web graphs and social networks.

**Strengths:**

The flow of the discussion is understandable and easy to read. Furthermore, a theoretical analysis of the effectiveness of the proposed method is provided, which guarantees the quality of the proposed graph reordering algorithm under certain assumptions. Theoretical guarantees are helpful because they cannot be demonstrated experimentally. Furthermore, the authors compared the proposed method with several graphs and existing experimental methods and emphasized its usefulness.

**Weaknesses:**

The paper has two major weaknesses.

1. Throughout the paper, it isn't easy to see the difference between the existing and proposed methods. It appears as if the proposed method incorporates randomness into the existing hash-based graph sorting algorithm. It should be clarified what method is used for the existing method that is the basis of the proposed method and how it differs from the proposed method. We believe this will further clarify the contributions of the proposed method and make the paper a good one.

2. Only small-world and scale-free graphs such as web graphs, social networks, and citation networks were used in the experiments. Although I understand that these graphs are commonly used in machine learning, the proposed HashOrder may be effective only for these highly central graphs. Therefore, it would be better to experiment with a broader range of graph data with different properties.

**Questions:**

The questions are related to what was mentioned in the Weaknesses section.

1. Can you briefly describe what kind of graph reordering algorithms are used in the existing methods based on the proposed method? Also, can you explain the difference between the proposed and existing methods?

2. Can you provide experimental results using other than small-world and scale-free graph datasets such as web graphs or social networks? Or can you discuss the performance of the proposed method on them?

---

> ### Author Response · Authors · 2023-11-22
>
> We thank the reviewer for recognizing the strengths of the paper in terms of the presentation, the theoretical support, and the empirical effectiveness of our proposed method. We address the reviewer’s concerns and questions below, and we are happy to add these discussions to the final paper. If the reviewer's concerns are sufficiently answered, we urge the reviewer to reconsider the score. If not, we would be happy to answer further clarifications.
>
> **1. Difference between existing methods and our proposed method.**
>
> Sorry for the confusion--there is no existing hash-based graph sorting algorithm. Our proposed method is a first of its kind probabilistic approach to the reordering problem, which is completely different from the existing methods. The existing methods can be categorized as follows 1) greedy optimization-based methods, such as GOrder 2) graph traversal-based methods, such as RCM 3) degree sorting-based methods, such as SortIn, SortOut, HubCluster, HubSort, and DBG. In contrast to these methods, HashOrder optimizes the fitness of all nodes simultaneously (unlike greedy and graph traversal-based methods) by leveraging randomized hashing and takes the higher order neighborhood into account (unlike degree sorting-based methods).
>
> **2. Can you briefly describe what kind of graph reordering algorithms are used in the existing methods based on the proposed method?**
>
> We apologize for not understanding this question. Could you please further elaborate on the question?
>
> **3. Experiments using datasets other than small-world and scale-free graphs.**
>
> Most real-world graphs exhibit small-world and power-law properties. Hence we study random graphs generated from Erdős-Rényi (ER) and Barabási-Albert (BA) models. ER is a well-studied scale-free random network model, while a BA network exhibits preferential attachment. We compare the runtime of one iteration of PageRank averaged over 5 runs on random graphs with different configurations, with n as the number of nodes and m as the number of edges. HashOrder achieves higher speedups than GOrder across different random graphs.
>
> |                                 | Original | GOrder (Speedup) | HashOrder (Speedup) |
> |---------------------------------|----------|------------------|---------------------|
> | Erdős-Rényi (n=100k, m=1M)      | 0.0866   | 0.0800 (7.68%)   | 0.0775 (10.54%)     |
> | Erdős-Rényi (n=100k, m=10M)     | 0.2613   | 0.2396 (8.29%)   | 0.2376 (9.06%)      |
> | Barabási-Albert (n=100k, m=1M)  | 0.1128   | 0.1021 (9.47%)   | 0.1014 (10.05%)     |
> | Barabási-Albert (n=100k, m=10M) | 0.3361   | 0.3038 (9.63%)   | 0.3028 (9.91%)      |

---

> > ### Comment · Reviewer_EBBV · 2023-11-23
> >
> > I think the following question was unclear, so I will change the content of the question.
> >
> > >2. Can you briefly describe what kind of graph reordering algorithms are used in the existing methods based on the proposed method?
> >
> > Regarding the Jaccard similarity and hashing methods proposed in this study, I couldn't find significant differences from the existing papers [1] [2]. There might be differences in the reordering of rows and graphs, but these do not appear to be major differences.
> >
> > [1] Jiang et al. (2020)
> > [2] Huang et al. (2021)
> >
> > For the next question, I assume that nodes are stored contiguously in memory. However, considering the size of the graph, it's safe to assume that the data size is much larger than the cache line size. When reading node information from a neighboring list, if we regroup similar nodes within the list, isn't there a lack of spatial locality? (Because they are likely to be in different cache lines). Therefore, it's unclear how reordering improves data reading performance in such a scenario.
> >
> > Regarding graph data from social networks, it is known to exhibit small-world properties. However, in contrast, graph data from transportation networks (like railways or roads) are composed of edges connecting geographically close nodes, leading to a larger graph diameter, and generally do not exhibit small-world properties. Thus, most real-world graphs do not show small-world and power-law characteristics. Of course, graph data from social networks is practically effective, but the dependence on the structure of graph data cannot be denied.
> >
> > In summary, I think there are challenges in differentiating from existing research. Also, since the performance of HashOrder and GOrder clearly depends on the characteristics of the graph, it would be better to explicitly state the graph characteristics where this algorithm is effective and discuss it.

---

> > > ### Author Response · Authors · 2023-11-23
> > >
> > > We understand the algorithmic synergies between our proposal and the works pointed out by the reviewer. We regret that the reviewer feels that our work is not novel. We believe we make the following contributions,
> > > 1. To the best of our knowledge, we are the first to apply LSH based technique in the domain of graph reordering for graph analysis problems.
> > > 2. We provide extensive evaluation of the method on datasets consistent with the graph reordering literature.
> > > 3. We bring the power of fast reordering of graph storage using hashing to the Graph Community.
> > >
> > > **How does reordering improve data reading performance when data size is larger than cache size?**
> > >
> > > Graph reordering not only benefits cache-level memory access performance, but also benefits performance of higher-order memory units such as pages at different levels.

---

### Official Review · Reviewer_KkKE · 2023-10-29

**Soundness:** 3 good
**Presentation:** 3 good
**Contribution:** 2 fair
**Rating:** 5
**Confidence:** 3

**Summary:**

This paper proposes an efficient graph reordering algorithm based on randomized hashing: HashOrder. Detailly, the authors propose a probabilistic algorithm for high-quality ordering, which is lightweight and parallelizable. Evaluations on graph processing workloads and GNN data loaders show that the proposed HashOrder outperforms the existing state-of-the-art method with considerable speedup.

**Strengths:**

1. The authors proposed an efficient graph reordering algorithm based on randomized hashing to improve graph algorithms. They also introduce a probabilistic perspective to demonstrate the advantages over previous algorithms. For the in-bucket ordering operation, they employ neighbor grouping and degree sorting techniques and analyze the ablation study for different numbers of hops k and threads to show the scalability of HashOrder.

2. The authors analyze the reordering time and overall execution speedup for graph algorithms. The proposed HashOrder achieves a better tradeoff between reordering consumption and reordered data quality.

**Weaknesses:**

1. novelty of this paper is limited. As admitted by the authors, LSH has been used in similar problems for graph reordering.

2. There is not much "machine learning" in this paper. While it targets GNNs, the paper is more of a system work in my opinion. I am not sure ICLR is the proper venue for it.

**Questions:**

1. The proposed HashOrder algorithm seems to be a CPU algorithm, since it is parallelizable, can it run on GPU?

---

> ### Author Response · Authors · 2023-11-22
>
> We thank the reviewer for recognizing the strengths of the paper in terms of the effectiveness of the proposed method. We address the reviewer's concerns and questions below. If the reviewer's concerns are sufficiently answered, we urge the reviewer to reconsider their score. If not, we would be happy to answer further clarifications.
>
> **1. LSH has been used in similar problems for graph reordering.**
>
> To the best of our knowledge, our proposed method is the first probabilistic approach to the reordering problem, which is completely different from existing methods. The existing methods can be categorized as follows 1) greedy optimization-based methods, such as GOrder 2) graph traversal-based methods, such as RCM 3) degree sorting-based methods, such as SortIn, SortOut, HubCluster, HubSort, and DBG. In contrast to these methods, HashOrder optimizes the fitness of all nodes simultaneously (unlike greedy and graph traversal-based methods) by leveraging randomized hashing and takes the higher order neighborhood into account (unlike degree sorting-based methods). LSH has been used for problems such as sparse-dense matrix multiplication, task scheduling, and load balancing, which are distinct problems from graph reordering and require dedicated solutions.
>
> **2. Can HashOrder run on GPU?**
>
> Yes, HashOrder can run on GPU since it conforms to the SIMD paradigm. The building blocks of the HashOrder algorithm are message passing with minimum aggregation and sorting, which are parallelizable operations on the GPU.

---

> > ### Comment · Reviewer_KkKE · 2023-11-23
> >
> > Thank you for the response. Could you explain why graph reordering works for GNN data loading even without spatial locality in the cache? The explanation given to Reviewer EBBV does not make sense to me. I think spatial locality in cache lines is the basic assumption for graph reordering algorithms such as GOrder to improve performance. The paper does not give details of how the data are loaded. Are they loaded from GPU memory or CPU memory? What data loading mechanism is used?

---

> ### Author Response · Authors · 2023-11-23
>
> We perform experiments for two data loading mechanisms (1) when data loading happens on CPUs and (2) when data loading happens via UVA mechanism on GPUs (see figure 4).  In both cases, even if the data per node is larger than cache-size, the colocation of data for two nodes which are accessed together inside a page helps.  We explain this below,
> 1. On GPU via UVA : In this case there will be a page miss when GPU accesses a node data from a page not previously accessed.
> 2. On CPU : In this case there will be a TLB miss when CPU accesses a node data from a page which is not present in the TLB.

---

> > ### Comment · Reviewer_KkKE · 2023-11-23
> >
> > Thanks for the explanation. I think it makes sense if the data are fetched in pages. However, I don't think GNN data loading is a good application for evaluating your technique. GPU has more efficient data loading mechanisms (zero-copy and DMA) that are more suitable for loading non-contiguous data. There has been solid work on this problem [1][2]. It is unclear whether the proposed technique in this paper would benefit if data are fetched with fine-grain mechanisms. I would suggest the authors tone down the applicability and focus on graph processing workloads.
> >
> >
> > [1] Large Graph Convolutional Network Training with GPU-Oriented Data Communication Architecture. VLDB'21.
> > [2] UGACHE: A Unified GPU Cache for Embedding-based Deep Learning. SOSP'23

---

> > > ### Author Response · Authors · 2023-11-23
> > >
> > > We acknowledge the opinion of the reviewer on GNN experiments. In our overall opinion, the paper tackles the graph reordering problem which is known to be a hard and impactful problem for many graph processing systems including ML systems. The hashing based method we propose is general and is widely applicable across a wide array of graph analysis algorithms including ML workflows. The generality, ease of implementation, fast reordering times and the high quality of ordering on a wide range of graphs (small world, scale-free, etc), as validated empirically, makes HashOrder a significant contribution.

---

### Official Review · Reviewer_L3q7 · 2023-10-31

**Soundness:** 3 good
**Presentation:** 3 good
**Contribution:** 3 good
**Rating:** 6
**Confidence:** 3

**Summary:**

This paper proposes to use HashOrder for remapping graph ids, with the goal of improving cache efficiency. The intuition is that cache utilization can be improved by placing neighbors that are frequently co-accessed together close in memory and that in-neighborhood intersection has direct connections to the cache efficiency metric. The authors propose to leverage minHash to compute LSH codes and then within each bucket, nodes are sorted by neighbor grouping or degree. Experiments are conducted.

**Strengths:**

1. The idea of using LSH ordering is interesting and cool.
2. The paper is easy to read.
3. The authors have conducted quite extensive experiments.

**Weaknesses:**

1. My main concern lies in the experimental results.
-- It does not seem to have significant improvements compared to Gorder in Figure 3.
-- It is better to report median speedup, instead of "Upto xx". The median speedup seems to be limited? and some are experiencing degration, e.g., in Figure 4.
-- It would be good to show end2end time for GNN training, instead of only reporting GNN data loader time in Figure 4.
2. It would be good if you can measure cache hit in the experiments.
3. Will reordering lead to other side effects? E.g., GNN training time and accuracy?

**Questions:**

1. why there exist blanks in Figure 4, e.g., graphsaint + speedup panel?
2. why choose the number of hash functions l = 2? will this affect the result?

---

> ### Author Response · Authors · 2023-11-22
>
> We thank the reviewer for recognizing the strengths of the paper in terms of the interestingness of the idea, the presentation of the paper, and the comprehensive evaluations of the proposed method. We respond to the reviewer’s concerns as follows.
>
> **1. Does not seem to have significant improvement compared to GOrder.**
>
> Our proposed method outperforms or is competitive with GOrder wrt downstream application runtime, while having significantly lower computational overhead. As shown in Figure 2, HashOrder surpasses GOrder in terms of ordering quality on average across all tasks, while being 592x (on average) more efficient in reordering.
>
> **2. Reporting median speedup/limited median speedup?**
>
> We see significant average speedups across tasks in downstream applications. We updated the paper to report the average speedup across datasets in the abstract. HashOrder speeds up PageRank by 1.44x and GNN data loaders by 1.93x on average.
>
> **3. Show end-to-end speedup for GNN training.**
>
> We present additional experimental results on end-to-end GNN training time. We train a 3-layer GCN on ogbn-papers100M using the MultiLayerFullNeighborSampler with different reordering algorithms, and measure the data loading and training time per batch. HashOrder outperforms RCM and achieves up to 36.55% speedup on end-to-end GNN training. We will add the results to the final paper.
>
> |                | Data Loading Time (Speedup) | Training Time (Speedup) | Total Time (Speedup) |
> |----------------|-----------------------------|-------------------------|----------------------|
> | Original Order | 4.971s                      | 1.245s                  | 6.216s               |
> | RCM            | 3.284s (33.94%)             | 1.053s (15.42%)         | 4.337s (30.23%)      |
> | HashOrder      | 2.952s (40.62%)             | 0.992s (20.32%)         | 3.944s (36.55%)      |
>
> **4. Measure cache hits in the experiments.**
>
> We have already included cache hit rates in Table 3 and 4 in the appendix.
>
> **5. Will reordering lead to other side effects?**
>
> We do not expect any negative side effects from reordering. Reordering improves the locality of the data layout so it accelerates training and inference. Accuracy is not affected as the graph remains the same with better memory layout.
>
> **6. Blanks in Figure 4.**
>
> Blanks in Figure 4 means the experiments are not feasible for those settings due to the inherent limitations of the GNN library. GraphSAINT sampler fails on ogbn-papers100M since torch.multinomial does not support a number of categories beyond 2^24. Full neighbor sampling using UVA fails on ogbn-products and ogbn-papers100M due to GPU memory limitations. We will improve the figure to clarify this.
>
> **7. Why l=2?**
>
> Empirically, we found l=2 marginally improves the ordering quality over l=1 (0.27% and 0.19% higher relative speedup). l=4 negatively impacts the ordering quality (7.89% and 1.46% lower relative speedup than 2 hashes), likely due to diminishing probability of collision.
>
> | PageRank run time in seconds (relative speedup) |                |                |                |
> |-------------------------------------------------|----------------|----------------|----------------|
> | Dataset                                         | l=1            | l=2          | l=4          |
> | web-Google                                      | 2.706 (57.68%) | 2.689 (57.95%) | 3.197 (50.06%) |
> | web-Stanford                                    | 0.830 (39.36%) | 0.827 (39.55%) | 0.847 (38.09%) |

---

### Official Review · Reviewer_3s87 · 2023-10-31

**Soundness:** 2 fair
**Presentation:** 3 good
**Contribution:** 3 good
**Rating:** 5
**Confidence:** 4

**Summary:**

The paper presents a row ordering strategy for sparse matrices based on locality aware hashing. Theoretical results are provided to show that this scheme is near-optimal for well-separable sparse matrices. The experimental evaluation considers the speed-up obtained by the reordering scheme for various graph work-loads, with other reordering schemes as baselines. Reordering time and parallelization are also considered.

**Strengths:**

+ The approach in the paper makes a lot of sense and appears to be new. While hashing has been used in many works on efficient sparse matrix computations, I was not able to find prior references that specifically consider locality-aware hashing of sparse matrix rows to optimize for locality.
 + The experimental results are promising, showing that the proposed reordering scheme outperforms several common alternative reordering methods.
 + The paper is generally fairly well-written and organized.

**Weaknesses:**

- I don't follow the structure of the theoretical results. Lemma 1 assumes existence of a hash family that is never proven, then Theorem 1 builds on Lemma 1, but drops the assumption that Lemma 1 had. This seems incorrect.
 - The paper misses what I would consider a seminal work that is also most-closely related to the approach in the paper
     Saad, Yousef. "Finding exact and approximate block structures for ILU preconditioning." SIAM Journal on Scientific Computing 24.4 (2003): 1107-1123.
   This paper aims to reoder similar rows of a sparse matrix together by hashing and by cosine comparison (similarity in nonzeros), as well as a hybrid method. These methods are different from that in the paper but obviously closely related and warrant discussion as well as experimental comparison.
 - The experimental results are based only on the authors' implementation. Some comparisons to existing libraries would be helpful in gauging whether the timings are competitive.
 - While the implementations are not fully described, the workloads largely seem to be all based on repeated SpMV and SpMM. It would be clearer to evaluate the reorderings for efficiency of those two basic kernels. The consideration of end-applications related to GNNs is secondary (they are dominated by SpMM), and seems motivated by the choice of publication venue.
 - Building on the prior point, I think there would be more expert reviewers, interest, and appropriate feedback for this type of paper at high-performance computing, scientific computing, and parallel computing publication venues such as Supercomputing, SPAA, IPDPS, SISC, etc., as these communities extensively study optimization of graph and sparse matrix primitives.

Overall, the paper presents an interesting and promising row-reordering scheme for sparse matrix products. However, the theoretical results seem to contain some errors or are imprecise. The experimental results are reasonable. I would be supportive of publication of this work if the theoretical results are corrected / claims are made more modest (assumptions provided) as appropriate. Due to the presence of issues with the theoretical result, and as ICLR seems a suboptimal publication venue to me for this type of work, I would lean toward rejection.

**Questions:**

Please provide clarification or planned revision regarding concerns of in the theoretical results.

---

> ### Author Response · Authors · 2023-11-22
>
> We thank the reviewer for recognizing the strengths of the paper in terms of the novelty of the idea and the effectiveness of the proposed method. We address the reviewer's concerns and questions below. The reviewer asked for two sets of additional experiments. Given the time constraints and the other experiments we had been working on, we have not been able to run them, but we will add them to the final version of the paper. If the reviewer's concerns are sufficiently answered, we urge the reviewer to reconsider their score. If not, we would be happy to answer further clarifications.
>
> **1. Structure of the theoretical results.**
>
> Sorry for the confusion. For Jaccard similarity which we use in HashOrder, the hash family assumed in Lemma 1 always exists since for any given $\alpha$ and $R$ such that $R > \alpha$, MinHash is a $(R, \alpha, R, \alpha)$-sensitive family. We have improved the presentation of the definitions and theorems (marked in blue) in the paper, but the theoretical result remains unchanged.
>
> **2. Discussion on "Finding exact and approximate block structures for ILU preconditioning."**
>
> The paper by Saad, Yousef explores reordering sparse matrices to form block structures for accelerating block ILU factorization. Although it focuses on sparse matrix reordering, to the best of our understanding, its objective is to achieve compression of the original matrix and reducing the number of fill-ins, which is different from the graph reordering objective of reducing the number cache misses. The two reordering problems are distinct from each other and require dedicated solutions. We will add the discussion and the experiments to the final paper.
>
> **3. The experimental results are based only on the authors' implementation.**
>
> In fact, we do use official implementations as much as possible. Our implementation extends the C implementation used in [2], which contains the official implementation of methods such as DBG, HubSort, HubCluster, SortIn and SortOut. As we have pointed out in the paper, we use the Ligra framework for running graph processing workloads and DGL for GNN workloads. We use the official implementation of RCM from the scipy package for the GNN experiments. Also, our experimental findings on reordering time and speedup of different methods are consistent with previous works.
>
> **4. The workloads largely seem to be based on repeated SpMV and SpMM.**
>
> While many workloads are related to SpMV/SpMM kernels, the range of graph analysis tasks are wide and many are difficult if not impossible to be reduced to SpMV or SpMM. For example, graph-based near-neighbor-search systems can be accelerated through reordering [1], but their searching process cannot be easily reduced to SpMV or SpMM. Moreover, many GNN data samplers cannot be efficiently implemented through SpMV or SpMM. In our paper, we evaluate a wide range of representative graph workloads, which are consistent with previous works [2]. Having said that, we are still happy to perform additional experiments and report them in the final paper.
>
> References
>
> [1] Coleman, Benjamin, et al. "Graph Reordering for Cache-Efficient Near Neighbor Search." Advances in Neural Information Processing Systems 35 (2022): 38488-38500.
>
> [2] P. Faldu, J. Diamond and B. Grot, "A Closer Look at Lightweight Graph Reordering," 2019 IEEE International Symposium on Workload Characterization (IISWC), Orlando, FL, USA, 2019, pp. 1-13, doi: 10.1109/IISWC47752.2019.9041948.

---

### Official Review · Reviewer_M1A7 · 2023-11-01

**Soundness:** 2 fair
**Presentation:** 2 fair
**Contribution:** 2 fair
**Rating:** 6
**Confidence:** 3

**Summary:**

This paper introduces an efficient and effective graph reordering algorithm utilizing node clustering based on randomized hashing. It theoretically shows that the orderings ensure some quality guarantees under clustering assumptions. The experiments verify that the proposal is efficient and effective.

**Strengths:**

S1. The experiments verify the effectiveness of the proposal under several representative graph analysis tasks.

S2. The proposal provides a theoretical guarantee for the quality of graph ordering.

**Weaknesses:**

W1. Regarding the evaluation of GNN data loading, the purpose is unclear. The evaluation should focus more on practical performance aspects of GNN, such as training time and inference time, Indeed, [1] conducted various experiments on typical GNN methods, such as GCN and GIN, using various graph analysis frameworks like DGL and PyG, to assess their training and inference times.

W2. Insufficient comparison with related techniques.
- The proposal performs node clustering using Minhash and parallelization. In fact, Rabbit order (Arai et al., 2016) shares a similar design concept, as it conducts node clustering using a modularity-based method and also incorporates parallelization. Hence, the authors should offer a comprehensive comparison between the proposal and Rabbit order, including performance experiments.
- The definition of fitness is slightly extended from the one introduced in GO algorithm (Wei et al., 2016), so novelty is relatively weak.

W3. Theorem 1 is founded on the cluster quality (Definition 1). However, it does not provide the size of \eplsion for Minhash, which is used in the proposal.

W4. It would be beneficial to include an end-to-end evaluation that encompasses both reordering and graph analysis.

**Questions:**

We would appreciate it if the authors could provide us with feedback regarding the points raised in W1-W4.

---

> ### Author Response · Authors · 2023-11-22
>
> We thank the reviewer for recognizing the strengths of the paper in terms of the empirical effectiveness and the theoretical support of the proposed method. We address the reviewer's concerns and questions below. If the reviewer's concerns are sufficiently answered, we urge the reviewer to reconsider their score. If not, we would be happy to answer further clarifications.
>
> **1. Evaluations on GNN training and inference time.**
>
> We present additional experimental results on end-to-end GNN training time. We train a 3-layer GCN on ogbn-papers100M using the MultiLayerFullNeighborSampler with different reordering algorithms in DGL framework, and measure the data loading and training time per batch. HashOrder outperforms RCM and achieves up to 36.55% speedup on end-to-end GNN training.
>
> |                | Data Loading Time (Speedup) | Training Time (Speedup) | Total Time (Speedup) |
> |----------------|-----------------------------|-------------------------|----------------------|
> | Original Order | 4.971s                      | 1.245s                  | 6.216s               |
> | RCM            | 3.284s (33.94%)             | 1.053s (15.42%)         | 4.337s (30.23%)      |
> | HashOrder      | 2.952s (40.62%)             | 0.992s (20.32%)         | 3.944s (36.55%)      |
>
> **2. Comparison with Rabbit Order.**
>
> We compare HashOrder with Rabbit Order on the efficiency of solving the SSSP (single-source shortest path) problem. For Rabbit Order, we use the official implementation from GitHub. The table below shows the relative speedup of different algorithms. HashOrder achieves greater speedups than Rabbit Order and GOrder. Rabbit Order generally performs worse than GOrder, which is consistent with the findings in [1].
>
> |             | Rabbit Order | GOrder | HashOrder |
> |-------------|--------------|--------|-----------|
> | web-Google  | 33.62%       | 38.14% | 42.27%    |
> | cit-Patents | 25.43%       | 27.92% | 36.75%    |
>
> **3. The definition of fitness is extended from the GO algorithm.**
>
> The definition of fitness in GO has drawbacks. It considers a sliding window and evaluates pairs of nodes inside the sliding window, and thus includes pairs which might lie in different cache lines/pages. However, in reality, evaluating pairs of nodes exclusively within a cache line/page better reflects the hardware. We fix this issue in our fitness definition. Additionally, we also generalize the inner product (neighbourhood intersection) used by GO to measure pairwise fitness. We find that our formulation allows for other metrics such as Jaccard similarity which can lead to efficient algorithms such as HashOrder.
>
>
> **4. The size of epsilon in the proposal.**
>
> We have reformulated the theorems for ease of reading introducing the definition of $(S,\alpha, \beta)$ clusterable graphs and theorem 1 for such graphs. The value of $\epsilon$ (reworded as $\beta$ in new formulation) is a property of the graph and not the hash function. It affects the quality of ordering, as mentioned in theorem 1, for all similarities and their corresponding hash families $(S, \mathcal{H})$ including Jaccard similarity and Minhash family of hash functions.
>
> References
>
> [1] M. Koohi Esfahani, P. Kilpatrick and H. Vandierendonck, "Locality Analysis of Graph Reordering Algorithms," 2021 IEEE International Symposium on Workload Characterization (IISWC), Storrs, CT, USA, 2021, pp. 101-112, doi: 10.1109/IISWC53511.2021.00020.

---

> > ### Comment · Reviewer_M1A7 · 2023-11-23
> >
> > Thanks for the detailed responses. Although the benefit of the proposal over to SOTA is not so large, the additional experiments actually prove a concrete benefit at the end2end evaluation setting (in particular, the total time of web-Google is great), so I raised the score to 6: marginally above the acceptance threshold.
> > Again, the comparison with Rabbit Order should be included in the new experiments.

---

> ### Author Response · Authors · 2023-11-22
>
> **5. End-to-end evaluation that encompasses both reordering and graph analysis.**
>
> In real-world industry applications, graph analysis can be prohibitively expensive and [3] shows that even reordering with high overhead can benefit the end-to-end application run time. Even in our experiments with simple graph analysis, we found that HashOrder can consistently speedup the end-to-end run time of PageRank, which we present the results in the following table.
>
> |          |            | Wiki-links   |            |             | LiveJournal  |            |             | Pokec        |           |            | web-Google   |           |            | web-Stanford |          |            | cit-Patents  |          |             |
> |----------|------------|--------------|-----------:|------------:|--------------|-----------:|------------:|--------------|----------:|-----------:|--------------|----------:|-----------:|--------------|---------:|-----------:|--------------|---------:|------------:|
> |          |            | Reorder Time | Run Time   | Total Time  | Reorder Time | Run Time   | Total Time  | Reorder Time | Run Time  | Total Time | Reorder Time | Run Time  | Total Time | Reorder Time | Run Time | Total Time | Reorder Time | Run Time | Total Time  |
> | PageRank | DBG        |     0.160721 | 349.840633 |  350.001354 |     0.045862 |  200.65444 |  200.700302 |     0.015944 |  86.62113 |  86.637074 |     0.009877 | 15.960602 |  15.970479 |      0.00243 |   3.2405 |    3.24293 |     0.020491 | 5.334885 |    5.355376 |
> |          | GOrder     |  3245.607901 | 323.327933 | 3568.935834 |  1055.639553 | 171.952428 | 1227.591981 |   361.788567 | 79.999555 | 441.788122 |     5.653868 |  8.809337 |  14.463205 |     3.662504 | 2.463918 |   6.126422 |  3467.710272 | 5.793895 | 3473.504167 |
> |          | HubCluster |     0.148035 | 361.118788 |  361.266823 |     0.042016 | 194.890394 |   194.93241 |     0.015516 | 87.443333 |  87.458849 |     0.008445 | 17.573982 |  17.582427 |     0.002023 |  3.48898 |   3.491003 |     0.018954 | 5.577104 |    5.596058 |
> |          | HubSort    |     1.555896 | 378.913859 |  380.469755 |     0.513904 |  224.32543 |  224.839334 |     0.158907 | 90.378812 |  90.537719 |      0.08431 | 15.786593 |  15.870903 |     0.023579 | 3.223704 |   3.247283 |     0.178334 | 5.583217 |    5.761551 |
> |          | HashOrder  |    13.522054 | 304.888094 |  318.410148 |     4.715484 | 178.290475 |  183.005959 |     1.259555 | 81.726037 |  82.985592 |     0.337742 |   7.89198 |   8.229722 |     0.115728 | 2.340298 |   2.456026 |     0.945633 | 4.863815 |    5.809448 |
> |          | Original   |            0 | 382.557465 |  382.557465 |            0 | 191.497494 |  191.497494 |            0 | 88.278854 |  88.278854 |            0 | 19.677888 |  19.677888 |            0 |   3.6446 |     3.6446 |            0 | 5.916659 |    5.916659 |
> |          | RCM        |     9.455407 | 322.334798 |  331.790205 |     4.281879 | 193.203711 |   197.48559 |     1.691509 | 82.190626 |  83.882135 |     0.459709 |  9.762702 |  10.222411 |     0.158757 | 2.331306 |   2.490063 |      0.23346 | 5.315313 |    5.548773 |
> |          | SortIn     |      1.57715 | 396.897457 |  398.474607 |     0.510017 | 227.540957 |  228.050974 |     0.152713 | 91.759981 |  91.912694 |     0.081956 | 14.885591 |  14.967547 |     0.023141 | 2.963184 |   2.986325 |     0.175311 | 5.151845 |    5.327156 |
> |          | SortOut    |     1.544873 | 347.899902 |  349.444775 |     0.494126 | 226.379185 |  226.873311 |      0.15189 | 90.022815 |  90.174705 |     0.084075 | 14.943749 |  15.027824 |     0.023768 | 3.145041 |   3.168809 |     0.108394 | 5.213656 |     5.32205 |
>
> References
>
> [2] V. Balaji and B. Lucia, "When is Graph Reordering an Optimization? Studying the Effect of Lightweight Graph Reordering Across Applications and Input Graphs," 2018 IEEE International Symposium on Workload Characterization (IISWC), Raleigh, NC, USA, 2018, pp. 203-214, doi: 10.1109/IISWC.2018.8573478.
>
> [3] Wei, Hao, et al. "Speedup graph processing by graph ordering." Proceedings of the 2016 International Conference on Management of Data. 2016.

---

### Author Response · Authors · 2023-11-23

Some reviewers have concerns regarding whether this paper is a good fit for ICLR. We believe it is for the following reasons. The efficiency of graph analysis is of significant importance in the learning community. Our paper focuses on the efficiency aspect of graph processing systems including machine learning systems. Historically, "systems for ML" has been an integral part of ML conferences such as Neurips [5]. Also, graph reordering papers, such as [1], have been published in recent ML conferences. The results of our paper potentially have wide applicability in learning applications, including similarity search [2], graph learning [3], knowledge graph reasoning [4], etc.

References

[1] Coleman, Benjamin, et al. "Graph Reordering for Cache-Efficient Near Neighbor Search." Advances in Neural Information Processing Systems 35 (2022): 38488-38500.

[2] Morozov, Stanislav, and Artem Babenko. "Non-metric similarity graphs for maximum inner product search." Advances in Neural Information Processing Systems 31 (2018).

[3] Kipf, Thomas N., and Max Welling. "Semi-supervised classification with graph convolutional networks." arXiv preprint arXiv:1609.02907 (2016).

[4] Jiang, Jinhao, et al. "Unikgqa: Unified retrieval and reasoning for solving multi-hop question answering over knowledge graph." arXiv preprint arXiv:2212.00959 (2022).

[5] http://learningsys.org/nips18/

---

### Meta-Review · Area_Chair_LzTH · 2023-12-05

**Metareview:**

The paper introduces a new method for ordering input graphs to optimize computation efficiency. The new method is studied both from a theoretical and practical perspective.

The paper contains some new interesting ideas and the experimental analysis is promising although during the discussion phase few limitations have been highlighted:

- the method has not been tested on large scale real world network

- the method is a bit incremental when compared with previous work on graph ordering

Overall, the paper is a nice contribution but it is below the ICLR acceptance bar.

**Justification For Why Not Higher Score:**

- the method has not been tested on large scale real world network

- the method is a bit incremental when compared with previous work on graph ordering

**Justification For Why Not Lower Score:**

N / A

---

### Decision · Program_Chairs · 2024-01-16

Reject